# A Robust Scheduling Framework for Re-Manufacturing Activities of Turbine Blades

**Lei Liu** * and **Marcello Urgo**

Department of Mechanical Engineering, Politecnico di Milano, 20133 Milano, Italy; marcello.urgo@polimi.it
* Correspondence: lei.liu@polimi.it

**Abstract:** Refurbished products are gaining importance in many industrial sectors, specifically high-value products whose residual value is relevant and guarantee the economic viability of the re-manufacturing at an industrial level, e.g., turbine blades for power generation. In this paper, we address the robust scheduling scheme of re-manufacturing activities for turbine blades. Parts entering the process may have very different wear states or presence of defects. Thus, the repair process is affected by a significant degree of uncertainty. The paper investigates the uncertainties and discusses how they affect the scheduling performance of the re-manufacturing system. We then present a robust scheduling framework for the re-manufacturing scheduling strategies, policies, and methods. This framework is based on a wide variety of experimental and practical approaches in the re-manufacturing scheduling area, which will be a guideline for the planning and scheduling of re-manufacturing activities of turbine blades. A case study approach was adopted to examine how re-manufacturers design their scheduling strategies.

**Keywords:** turbine blades; re-manufacturing; uncertainty; robust scheduling

## 1. Introduction and Industrial Motivation

In the past few years, increasing attention has been devoted towards enhancing the sustainability of manufacturing processes by reducing the consumption of resources and key materials, energy consumption and environmental footprint, while also reducing costs and increasing competitiveness in the global market. Re-manufacturing, which can be defined as "the rebuilding of a product to specifications of the original manufactured product using a combination of reused, repaired and new parts" [1], is a form of product recovery process entailing the repair or replacement of worn out components to obtain re-manufactured products with the same characteristics as new products. The re-manufacturing paradigm is aimed at supporting sustainability challenges in strategic manufacturing sectors for high-value products whose residual value is high, such as aeronautics, automotive, electronics, consumer goods, and mechatronics [2].

Re-manufacturing processes, compared to the original manufacturing processes, entail higher degrees of uncertainty, complexity and dynamics, due to the unpredictable and variable conditions of the used parts to be processed. This significantly affects process and production planning, as well as the requirements driving the design of systems operating re-manufacturing activities. Thus, low production efficiency, unstable product quality, frequent abnormal production accidents and high product rework rates are typical characteristics of re-manufacturing systems [3]. To address these operating scenarios, smart approaches are needed to match production management and control decisions with the updated state of production processes, resources and requirements.

Industry 4.0, which is one of the most trending topics in manufacturing area, considers smart manufacturing as its central element, and relies on the adoption of digital technologies such as Internet of Things (IoT), cloud services, big data and analytics, to gather data in real time and to analyze it, providing useful information to the manufacturing system [4].

To hedge against unexpected events in the manufacturing system, predictive and reactive approaches can take advantage of advanced sensor technologies providing monitoring capability and artificial intelligence supporting analyses and decisions [5]. Taking advantage of the described scenario, smart management approaches for re-manufacturing systems are required to support management, planning and scheduling, within the circular economy paradigm [2].

Grounding on this perspective, this paper focuses on scheduling approaches for re-manufacturing activities, able to cope with uncertainty affecting both processing times and steps. The aim is to devise a robust schedule to mitigate the impact of these uncertain events. The industrial application addressed is gas turbine blade, an extremely complex and high-value product.

The paper is organized as follows: Section 2 reviews relevant literature; Section 3 describes the addressed re-manufacturing environment entails the characteristics and operating environment of turbine blades, re-manufacturing process and associated uncertainties; Section 4 presents the proposed robust scheduling framework, while a case study is reported in Section 5. Finally, Section 6 provides final considerations and conclusions.

## 2. Literature Review

Re-manufacturing, as one of the most important sustainable economy paradigms, has drawn a lot of attention due to its advantages on cost-effectiveness, energy-saving and emission-reduction. Tolio et al. [2] revises system level problems, methods and tools to support the re-manufactuing paradigm and highlights the main challenges and opportunities towards a new generation of advanced re-manufacturing systems. Goodall et al. [6] reviews the work on tools and methods which have been developed to support the decision process of assessing and evaluating the viability of conducting re-manufacturing, and evaluate how they have met the requirements of the decision stage.

Turbine blades re-manufacturing environment is an emerging industrial sector where many advanced tools and technologies are proposed. Rickli et al. [7] described a framework for re-manufacturing systems that aims to take advantage of additive manufacturing processes to remanufacture end–of–life cores. Huang et al. [8] proposes a re-manufacturing scheme design method based on the incomplete reconstruction of used part information to solve the uncertain and highly personalized problems in re-manufacturing. Nevertheless, re-manufacturing environment for turbine blades is way more complicated than the traditional manufacturing environment, and it involves enormous decision-making practices due to the uncertainties incurred by the different demand, inventory, processing routes and times, which depend on the condition of the returned products [9]. However, among the literature on the tools and approaches for re-manufacturing environment, 48% focus on strategic-level and 34% on tactical-level, with only 5% focusing on operational-level, evenly, only 36% of the studies address uncertainties which is a highly important factor companies face in product recovery management [10,11].

Production planning and scheduling, which play important roles in the organization of manufacturing activities and directly affect the overall performance of manufacturing, is also extensively studied in the re-manufacturing environment. Morgan and Gagnon [12] classified the re-manufacturing scheduling into disassembly versus integrated scheduling, single versus multiple products scheduling and reviewed the relevant literature. Kin et al. [13] proposed a conceptual methodology for reconditioning process sequence planning based on the ranking of the defects and the precedence relationships which consider the criticality of the defects. Zhang et al. [14] addressed the re-manufacturing scheduling problem by adopting a simulation-based optimization framework and proposed a genetic algorithm to optimize two objective functions. Liu and Urgo [9] proposed a approximate branch and bound algorithm for 2-machine permutation flow shop scheduling re-manufacturing activities for the repair of turbine blades, with a value-at-risk performance investigated.

In the re-manufacturing of turbine blades, two main sources of uncertainty are processing routes and times, which depend on the condition of returned products. To our best



knowledge, no general framework exists to support the scheduling of re-manufacturing activities with uncertain processing routes and times.

### 3. Re-Manufacturing of Turbine Blades

Gas turbine (Figure 1), in which burning of an air-fuel mixture produces hot gases that spin a turbine to produce power, is one of the most widely-used power generating technologies. Turbine blades (Figure 2), whose individual price is close to a middle-class car (i.e., about ten-thousand euros), is one of the most important and expensive parts in a gas turbine. To maximize the performance of a gas turbine, they are constituted by multiple stages, each of them equipped with specially-designed turbine blades. According to the specific OEM (Original Equipment Manufacturer), the number of stages can vary, as an example, a F-class turbine [15] needs about 400 blades for 3 to 6 stages. Blades belonging to the same stage of the turbine are usually manufactured/re-manufactured in batch to guarantee homogeneous characteristics and the balancing of the portion of the rotor for the stage.

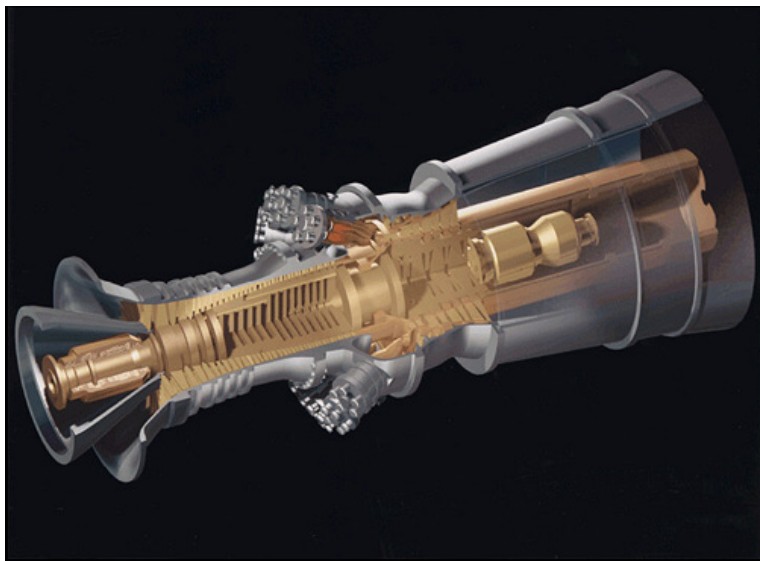

**Figure 1.** GE H-series power generation gas turbine [16].

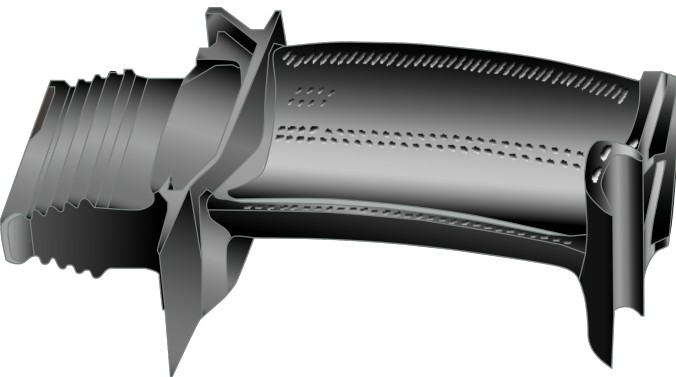

**Figure 2.** New turbine blade [17].

During the functioning of the turbine, blades are exposed to an extremely strenuous environment (high temperatures, stresses, vibration and corrosion) constituting a major cause for wear and failures of the blades (Figure 3) and, consequently, reduced energy conversion efficiency, and potentially disruptive failures of the whole turbine.

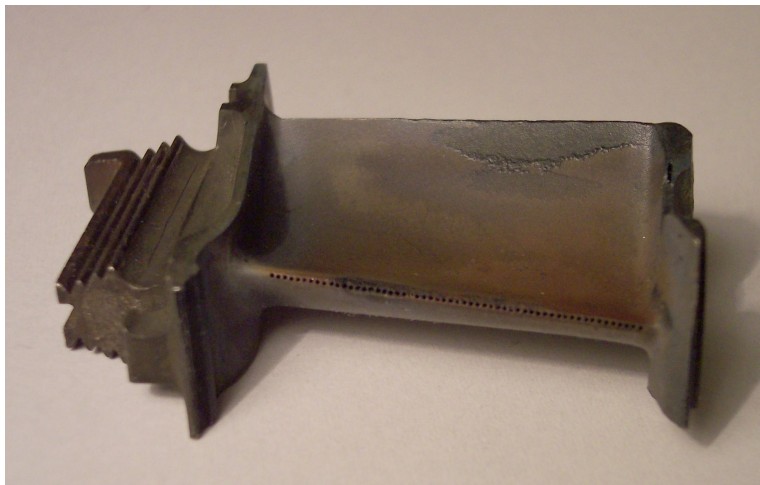

**Figure 3.** Damaged blades resulting from exposure to high temperature and stress [17].

Thus, planned maintenance operations are enforced for gas turbines, requiring all the blades in the rotor to be disassembled, inspected and, if needed, re-manufactured or replaced with new ones.

### 3.1. Re-Manufacturing Processes

Taking the economy cost into consideration, extending the service life of blades through re-manufacturing is a feasible and attractive choice, since their replacement with a new one is far more expensive.

The re-manufacturing of turbine blades generally involves multiple processes and technologies, e.g., visual inspection, non-destructive testing, machining, additive manufacturing, grinding, heat-treatments, coating, etc. The whole process can be summarized in the following main steps:

1.  Blades are disassembled at the production site and shipped to the OEM to be refurbished/renewed.
2.  Upon arrival at the OEM, blades are inspected to assess the severity of wear and damages. Blades that cannot be repaired are discarded and the supply/production of new blades is issued. On the contrary, for reparable blades, the re-manufacturing process starts, whose process parameters depends on the assessed damage level.
3.  The re-manufacturing process consists of removing the damaged (e.g., cracked) parts of the blades, restore the removed material through welding/additive technologies, machining/grinding the blade to reconstruct the desired shape. During the material removal phase, as well as at the end of the processing, non-destructive tests are operated to verify the complete removal of the defects.
4.  The blades undergo the definition of small-size features (e.g., internal cooling ducts, specific shapes in the terminal side or in the coupling with the rotor) and are successively coated with high-resistance materials.
5.  Finally, the blades belonging to the same stage are assembled together and balanced. Hence they are shipped back to the operating site of the turbine where they will be re-assembled and be ready to go into production again.

In this paper the focus is on the core part of the re-manufacturing process, i.e., only the phases operating the removal of the defects, the reconstruction of the original shape through an additive process, and the machining of the blade to obtain the desired final shape. While the blades go through these phases, due to the different and unpredictable degree of wear and severity of the damages on the parts, rework activities and non-destructive testing operations could be needed, as shown in Figure 4.

Thus, depending on their characteristics, wear and possible presence of damages, different blades will undergo different sequences of operations. Figure 4 presents an

example of the repairing process for two different batches of turbine blades. The process flow of the first batch of blades (i.e., stage 1) is described with a black arrow, while a blue dashed arrow is used for the second batch blades. $O_i$ denotes the operations belongs to defects removal, material additive, machining and NDT process steps respectively. Although both stage 1 and stage 2 blades undergo the four main process steps, the detailed operations could differ according to the specific stage they belong to.

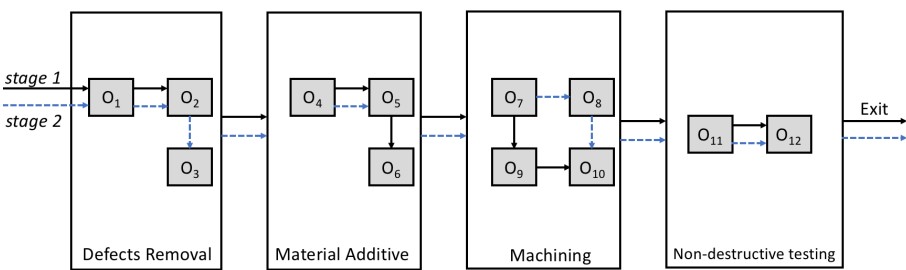

**Figure 4.** Re-manufacturing process example for two batches of turbine blades.

*3.2. Uncertainty in Re-Manufacturing Processes*

One of the peculiar requirements of re-manufacturing processes is the capability to cope with the intrinsic uncertainty affecting the parts and products to process. This drives the need for re-manufacturing plants able to operate in variable conditions with high degree of efficiency awhile guaranteeing the quality of the processed products [2,18].

The main sources of uncertainty to manage can be summarized in the following classes:

1.  A major source of uncertainty is embedded in the characteristics and volumes of the parts to be processed. Used parts arrives from different use-phase conditions and, thus, could be worn differently. Moreover, grounding on their characteristics, the number of parts whose re-manufacturing is feasible or attractive could be different, thus impacting on the actual volumes of parts to be re-manufactured.
2.  The re-manufacturing process plan could vary for different specimens of the same products. Thus, according to the characteristic of the item (wear, presence of damages, etc), different process steps arranged in different sequences could be needed. This impact on the management of the re-manufacturing system due to the need of implementing different routines among the available resources.
3.  Grounding on the characteristics of each specific part, re-manufacturing process can differ in terms of the characteristics and parameters of re-manufacturing operations. This a clear impact on the execution of the processes (forces, velocity, processing times, etc.)

The described sources of uncertainty also fits the case of re-manufacturing turbine blades. As stated in Section 3, blades are repaired in batches, consisting of a set of blades belonging to the same stage of the turbine. Blades belonging to different stages, as well as blades in different positions in the same blade, could be exposed too different stresses and environmental conditions and, thus, result in different degrees of wear. Moreover, after the initial inspection phase, a subset of the blades results not repairable and must be discarded. Thus, the number of blades to be processed as well as their characteristics are not known in advance.

Moreover, repairable blades with a higher degree of damages will likely entail longer processing times and different process parameters respect to the less severely damaged blades. The severity of the damages can be partially estimated in the initial inspection phase but, while the re-manufacturing process is operated, deviations from what initially estimated could emerge. These deviations can just affect the processing time of specific operation or, in some cases, require different sequence of operations to be executed, e.g., non-destructive testing and rework (Figure 5). Thus, processing times as well as the steps to be operated in the system entail a certain degree of uncertainty.

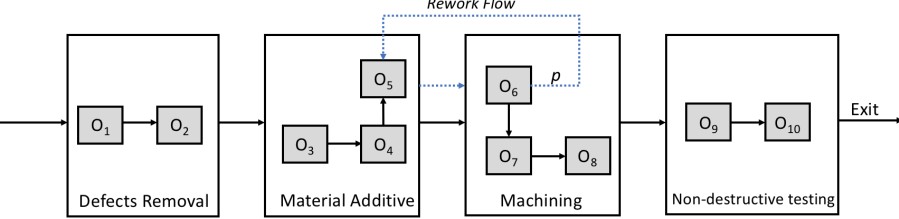

**Figure 5.** Repairing process example for a batch with rework.

The sources of uncertainty described above primarily affects the planning and scheduling of re-manufacturing operations, responsible of matching the requirements of the customers (delivery dates) as well as defining planned execution of operations matching the characteristics and constraints within the re-manufacturing system. Thus, it would not be possible to obtain a realistic schedule of re-manufacturing operations without being able of taking into consideration the complexity and uncertainty of the described re-manufacturing environment [19], entailing the need of the development and adoption of robust scheduling approaches [20,21].

## 4. Robust Scheduling Framework

Robust scheduling approaches focus on constructing preventive schedules to minimize the effects of disruptions on the performance measure and try to ensure that the predictive and realized schedules do not differ drastically, while maintaining a high level of schedule performance [20]. In this section, a resource constrained project scheduling (RCPSP) approach is used to formalize the re-manufacturing process of turbine blades, different modelling approaches are used to describe relevant sources of uncertainty, proper robustness measures are introduced to support the devising of robust schedules.

### 4.1. Shop Scheduling Model

Grounding on the description of the re-manufacturing process in Section 3.1, several batches of turbine blades are repaired in the same shop, different batches may need different sets of operations, sharing the same resource (e.g., workers, machines). The scheduling of batches of turbine blades can be modeled through a resource constrained project scheduling problem (RCPSP) with limited renewable resources [22]. The structure of the processes to be operated is defined by precedence relations among the activities, the associated processing times can be modeled through random variables to consider the associated uncertainty. The aim of the scheduling approach is the minimization of the makespan, supporting the optimized utilization of the resources [23].

Each activity represents a processing operation of a whole batch of blades in a specific process phase. Activities cannot be interrupted, hence, a non-preemptive schedule is pursued. The set of precedence relations are usually given as a directed acyclic graph, where nodes represent activities while an edge $(u, v)$ models a precedence relation enforcing $u$ to finish before $v$ is allowed to start. The graph contains two dummy activities, the source node S and the sink node T, modeling the start and finish of the whole set of activities. A graph representing the processing of $k$ batches of blades is reported in Figure 6. A set of batches $(1, 2, 3, \ldots, k)$ of turbine blades which come from different of customer orders and stages need to be re-manufactured, each horizontal chain from source node S to sink node T represents the processing steps of one batch of turbine blades. Different batches undergo different process steps while with many equal operations $(A, B, C, D, \ldots)$ consuming same resources.

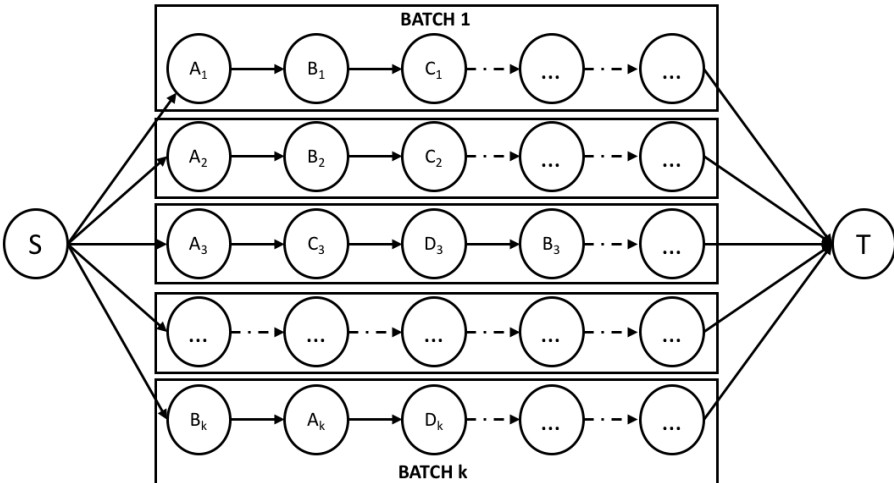

**Figure 6.** RCPSP model for re-manufacturing process of turbine blades.

The set of renewable resources represents production resources in the shop (e.g., experienced workers, machines). Each resource has a maximum capacity per time slot while each activity is associated to a requested amount of each resource for each time slot (greater or equal to zero). The dummy source and sink activities require no resource.

With respect to the objective to be pursued, the minimization of the makespan (i.e., the completion time of the dummy sink activity) is addressed in this paper. Nevertheless, pursuing a robust approach entails the need of addressing the uncertainty affecting the scheduling problem in the objective function. The following subsections are going to address this aspect.

### 4.2. Modeling Uncertainty

Grounding on the description of uncertainties in the re-manufacturing process of turbine blades in Section 3.2, different classes of unexpected events must be modeled. Specifically, with respect to turbine blades, the effects of these classes of uncertainties affects processing times and the sequence of process steps.

#### 4.2.1. Uncertainty Affecting Processing Times

The uncertainty affecting processing times can be modeled in different ways, according to the available amount of knowledge and data. The simpler approach is the definition of an interval so that the processing time of an operation $p_i$ can assume values within an interval $[p^{L_i}, p^{U_i}]$, i.e., $p_i \in [p^{L_i}, p^{U_i}]$. The probability distribution associated to the interval can be the uniform one [24].

Pursuing this approach, only the possible minimum and maximum values of the processing times should be determined, grounding on the available knowledge or on historical production data [25]. In the re-manufacturing of turbines, the blades are repaired in batches and the processing time uncertainty associated to a batch depends on two main factors as anticipated in Section 3.2:

1.  The number of blades in a batch depends on their level of damage. Extremely worn blades cannot be refurbished and must be discarded. A rejection rate table based on the damage level of each type of blade is defined. Thus, each batch of blades to be processed is assigned a level of damage (high, medium, low) sampled from a discrete distribution. Hence, the rejection rate is modeled as an uniform distribution defined by a lowest and highest rejection rates for that class of blades.
2.  The processing time of an operation on a single blade is modeled as an interval whose boundaries can be determined grounding on historical data. Thus, the distribution of the processing time of a batch of blades can be obtained through a convolution

operation between the distribution of the number of blades in the batch to be re-manufactured and the distribution of the processing time of a single blade.

If more detailed information and knowledge are available, uncertainty related to processing times can be modeled by independent random variables with an associated general probability distribution, either discrete or continuous [26]. Differently from the approach described above, additional hypotheses are required for the definition of the probability distributions, namely, their type and the associated parameters. The main rule here is trying to limit hypotheses that cannot be fully supported by the available data. Thus, simple distributions (e.g., triangular) are preferred [27].

### 4.2.2. Uncertainty Affecting the Process Steps

In re-manufacturing, the characteristics of the parts to be processed, e.g., their level of wear, could entail the need of operating different processes. A typical situation is the need of executing a rework.

The execution of rework activities can be modeled in different ways, grounding on the specific characteristic of the process and re-manufacturing environment as well as with respect to the available data. A first approach is incorporating possible rework activities in the standard process, thus, modeling the distribution of processing time to take into consideration the possibility of longer times due to the need to operate a rework. Although simple, this approach is clearly giving up in modeling the actual sequencing of operations. In fact, rework activities are usually operated after a verification of the standard process. Thus, they are usually operated at a later temporal stage. Putting work and rework operations in the same scheduled activity is not correctly modeling the fact that, in the meanwhile, other activities can be processed. A second approach is hypothesizing that all batches need to be reworked, which is a really common occurrence in the re-manufacturing of turbine blades. Thus, an additional set of activities is considered, for each batch of blades to be processed). These are then added to the original RCPSP introduced in Section 4.1, considering the associated processing times as well as precedence constraints.

Another approach considers that not all the possible rework activities are always operated. Thus, their occurrence is modeled through a Bernoulli distribution while the associate processing time is still defined by a probabilistic distribution, thus:

1.  A RCPSP model is defined considering both work and rework activities.
2.  The processing time of rework activities is obtained through the convolution of the occurrence and processing time distributions described above.

### *4.3. Robustness Measures*

A robust schedule, which is defined as a schedule that is insensitive to unforeseen disturbances [20], minimizes the effect of uncertainties on the primary performance measure of the schedule when implemented. Most of the robust scheduling approach consider the expected realized performance of the schedule, while it is really limited since minimizing the expected value fails in estimating the quality of the schedule in a stochastic point of view [28]. In this paper, we consider risk based robustness measures, specifically, the min-max regret, value-at-risk and conditional value-at-risk of the realized schedule are investigated.

### 4.3.1. Minimizing the Maximum Regret of the Objective Function

The min-max regret scheduling approach is a risk-aversion method focusing on the worst case scenarios, rather than considering all the possible realizations. It takes into account the regret, i.e., the deviation of an outcome from the best possible one in each specific scenario. Thus, this is not an absolute measure of performance of the solutions, but relative to the best available performance for a specific scenario [29].

The described re-manufacturing process of turbine blades with uncertain processing times and rework activities is modeled as a resource constrained project scheduling (RCPSP) problem (Section 4.1), grounding on the modeling of the uncertainty related to processing

times and rework activities as described in Section 4.2. For this class of problems, the optimization of the maximum regret can be addressed considering extreme scenarios only [30]. In the case under study, this is pursued considering extreme scenarios for the processing times $p_i$, i.e., with $p_i = p_i^{min}$ or $p_i = p_i^{max}$ for all $i$ and exploiting the algorithm proposed in [31].

### 4.3.2. Minimizing the Value-at-Risk/Conditional Value-at-Risk of the Objective Function

The minimization of the maximum regret has the advantage of only requiring knowledge about the extreme scenarios and, consequently, the simple knowledge of the extreme values of the domains for uncertain variable is enough. At the same time, focusing the optimization on worst-case scenarios, that may be unlikely to occur, tends to be too conservative. To cope with this limitation, different risk measures can be used to guide the optimization, namely, the value-at-risk (VaR) and the conditional value-at-risk (CVaR) [32]. These indicators take into consideration the whole distribution of uncertain variables and, thus, are able to consider the impact of uncertain events both in terms of their effect and occurrence probability. The use of both the VaR and CVaR was initiated in the financial area [32] but their popularity in robust scheduling area is rapidly increasing [21,33,34].

In the case of the deterministic RCPSP, a solution to the RCPSP is a schedule $s$, i.e., a vector of starting times $(s_0, s_1, \ldots, s_n, s_{n+1})$, with each activity duration is a constant. While for the scheduling problem under study, the decision maker does not know which exact information of activity duration, and yet a number of sequencing decisions need to be made. Hence, the execution of the project with uncertain activity durations is a dynamic decision process, and a schedule solution which denoted as a vector $x$, is a policy, which defines actions at the start of the project and at the completion times of activities. A vector of random variables $y = \{p_1, \ldots, p_n\}$ models the random processing times, governing by a probability measure $P$ on $Y$ and independent of scheduling decision $x$. The probability distribution of the makespan, $f_{C_{max}}(x, y)$, depends on the values of $x$ and $y$. For a given schedule $x$, the resulting cumulative density function (cdf) for the makespan is defined as:

$$F_{C_{max}}(x, \zeta) = P(f_{C_{max}}(x, y) \leq \zeta | x) \tag{1}$$

Then, the value-at-risk $\alpha$ ($VaR_\alpha$) of $C_{max}$, associated with a schedule decision $x$, denoted as $\zeta_\alpha(x)$, is defined according to the following:

$$\zeta_\alpha(x) = min\{\zeta | F_{C_{max}}(x, \zeta) \geq \alpha\} \tag{2}$$

Further, the $\alpha - CVaR$ of (1) associated to a schedule $x$ is the mean of the $\alpha-$tail distribution (3) [35].

$$F_X^\alpha(z) = \begin{cases} 0, & \text{when } z < \zeta_\alpha(X) \\ \frac{F_X(z) - \alpha}{1 - \alpha}, & \text{when } z \geq \zeta_\alpha(X) \end{cases} \tag{3}$$

By considering the proposed Equation (2), VaR can be specified as the risk measure on the random $f(x, y)$, and minimizing $\zeta$ corresponds to seeking the schedule solution with the smallest possible VaR measure for a specified $\alpha$ value, i.e., for confidence level $\alpha$, $VaR_\alpha$ is the $(1 - \alpha)$- quantile of the makespan distribution which is the largest value that ensures that the probability of obtaining a makespan less than this value is lower than $1 - \alpha$ [33].

CVaR at confidence level $\alpha$, i.e., $CVaR_\alpha$, is defined as the expected value of makespan smaller than the $(1 - \alpha)$-quantile of the probability distribution of makespan, i.e., $VaR_\alpha$ [36]. CVaR is the expected value of the makespan for the worst $\alpha\%$ cases with a value greater than the VaR.

The main difficulty in pursuing robustness through the minimization of the VaR/CVaR lies in the need to calculate the distribution of the objective function, e.g., the makespan. In scheduling problems affected by uncertainty, thus entail the capability of dealing with the correlation among all the possible paths in the network of activities (Section 4.1) [37,38]. To overcome this difficulty, a Markovian Activity Network(MAN) approach can be used to

support the analytical estimation of this distribution. Basic MANs require the processing times of the activities to follow exponential distributions [39] but can be extended to cope with general distributions, approximated by phase-type distributions [40]. Grounding on this, the distribution of objective function based on the completion times of the activities (e.g., the makespan), enables the use of risk measures to address robustness of scheduling.

## 5. Case Study

The proposed robust scheduling framework has been preliminary tested in an industrial environment to support the scheduling of the re-manufacturing activities for turbine blades. In this case study, four turbine stages are considered. Each stage consists of a set of identical blades that are thus processed as a batch. The steps of the repair process are the same for all the batches, i.e., defects removal, material additive, machining, and non-destructive testing (Section 3). Within these steps, 9–12 operations in total are executed, with some differences among the different types of batches. Moreover, 2/3 operations in the process may need a successive rework. The initial state of the turbine blades is modeled in terms of their damage level, defined for each stage grounding on historical data, reported in Table 1. Starting from the damage level, a rejection rate can be defined (the probability for a blade to be too damaged to be repaired), as well as the distribution of the processing times for rework operations (Section 4.2.1). Each activity is expected to require a single renewable resource from a resource set with four different resources representing machines and human workers involved.

**Table 1.** Damage level table.

| Damage Level | Rejection Rate LB (%) | Rejection Rate UB (%) |
|:---:|:---:|:---:|
| Heavy | 60 | 90 |
| Medium | 30 | 60 |
| Light | 0 | 30 |

We will show how to appropriately use the framework to provide relevant information to the decision-makers, and help them develop robust schedules in their turbine blades re-manufacturing plants.

A resource constrained project scheduling model (RCPSP) is used to formalize the described scheduling problem (Section 4.1). A total of four batches are considering, resulting in 41 activities to schedule (including the start and complete dummy activity). The objective of the approach is the minimization of the maximum regret of the makespan (Section 4.3.1). The processing time of each activity is modeled with an interval derived from historical data and convolution operations (Section 4.2.1). Rework activities are described through a Bernoulli distribution modeling the probability to occur (Section 4.2.2). Thus, the integrated uncertainty is described by a set of scenarios, obtained by considering the possible occurrences of processing times and reworks for all the activities. For each scenario, a processing time vector $[p_0, p_1, \dots, p_{40}]$ is used to describe the processing time for all the activities. Since only extreme scenarios need to be examined in the maximum regret minimization model [30], $2^{(41-2)}$ scenarios in total need to be evaluated in this case study, the scenario relaxation algorithm with accelerated convergences for robust resource-constrained project scheduling problem proposed in [23] has been adopted to support the definition of robust schedules.

To demonstrate the benefits of this robust scheduling framework, an alternative scheduling approach is used for comparison, namely, one only considering the expected values of processing times and a probability of 50% rework occurrence, ignoring the disturbance and volatility in the re-manufacturing process, is compared with the proposed framework.

Due to the large amount of possible scenarios, it is unfeasible to test the schedule obtained through the alternative approach on all of them. To overcome this limitation, a subset of the total number of scenarios is sampled and, for each of them, the performance

obtained with the schedule considered (i.e., the makespan). These values are then used to assess the value of the stochastic solution (VSS), i.e., the value of exploiting stochastic information to support a robust schedule. It is calculated as the average deviation between the performance of the robust approach and the one just considering expected values (Equation (4)) over the considered set of scenarios [41].

$$VSS = \frac{1}{|S|} \sum_{s \in S} (EVS_s - RVS_s) \tag{4}$$

In Equation (4), EVS denotes the value of the solution considering expected values of the stochastic variables, and RVS is the one of the robust schedule. Moreover, $|S|$ is the number of scenarios considered, and $s$ denotes a specific scenario in the set.

The proposed robust scheduling framework has been tested on 50 randomly generated problem instances. Each instance represents the re-manufacturing of a turbine with all its stages. Considering the results obtained for the whole set of instances, the relative value of the VSS is on average 5.4%. Since the absolute value of the VSS depends on the test instance, a scatter plot of the relative values of the VSS is presented in Figure 7, showing that for the vast majority of them, values are positive. This supports the value of the proposed robust scheduling framework in comparison with the expected value solution (EVS).

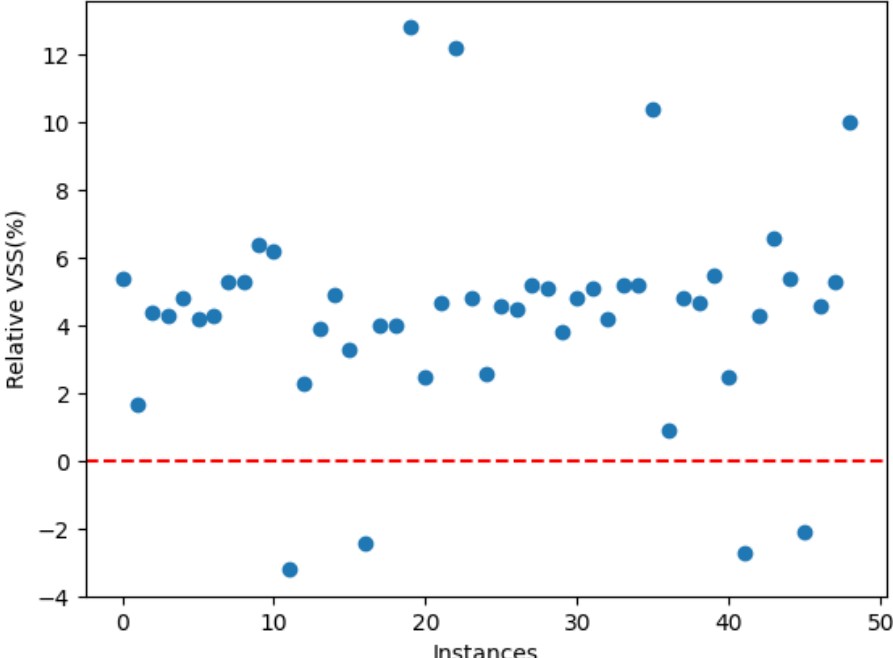

**Figure 7.** Relative value of VSS.

To further investigate the performance of the proposed robust approach, an instance has been selected, and the objective values of the two approaches with respect to 20 randomly generated scenarios. The results are shown in Figure 8 (left), showing that, for most of the scenarios, the robust scheduling framework performs better, i.e., leads to a smaller makespan. Furthermore, the results obtained in the described scenarios has been evaluated in terms of the difference between the makespans obtained. The results are reported in Figure 8 (right), showing that the proposed robust approach performs better on average (the 0.25-quantile of the difference EVS-RVS is positive). Moreover, the top whisker of the plot shows that, in extremely unfavorable cases, the protection provided by the robust schedule is extremely valuable.

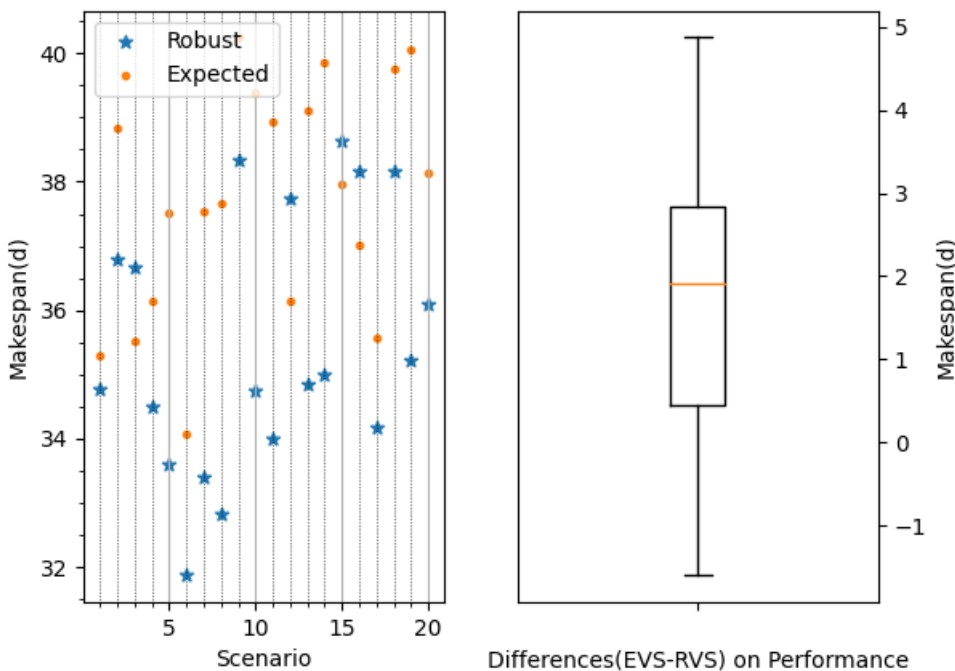

**Figure 8.** Performances under two approaches.

In addition, an extreme scenario has been analyzed, i.e., a worst-case scenario where all the processing times have the maximum possible value and all rework operations occur. The resulting execution of the robust schedule and the one obtained considering the expected values of uncertain variables are represented in Figure 9 where the blue one is the robust schedule and red the expected value one. The figure provides an aggregate representation, showing the aggregate processing of stages, and a detailed one, showing the detailed processing of all the re-manufacturing activities. It is clear from the Gantt in Figure 9 that, upon the occurrence of an extreme scenario, the robust schedule guarantees a smaller makespan.

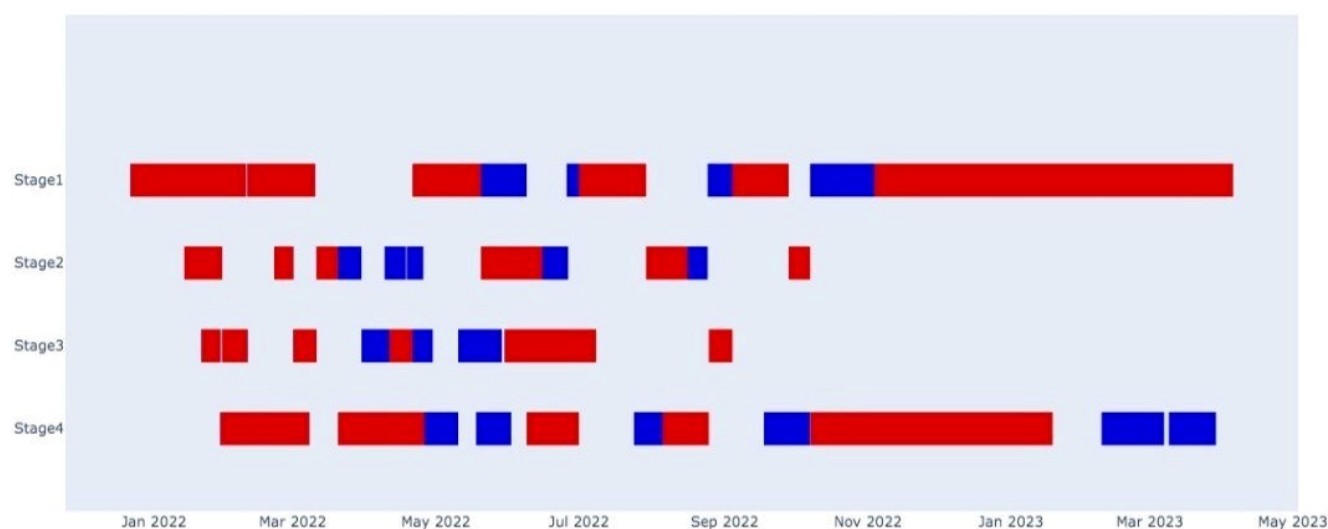

**Figure 9.** *Cont.*

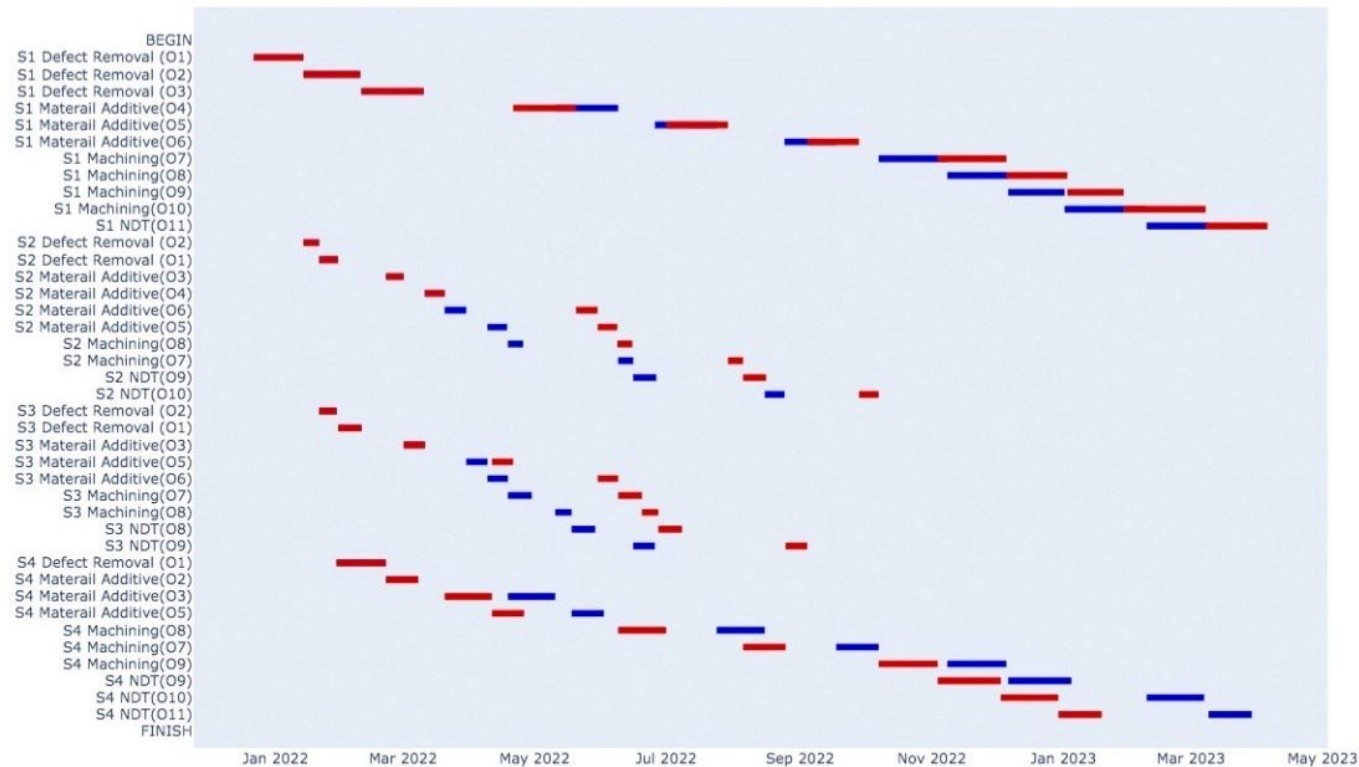

**Figure 9.** Gantt Chart.

## 6. Conclusions

In this paper, a robust scheduling framework is proposed to support re-manufacturing activities for gas turbine blades. The re-manufacturing problem and the relevant sources of uncertainty are described and formalized. To pursue the definition of robust schedules, different robustness measures, i.e., minimax regret and VaR/CVaR, are exploited. A case study, which can be a guideline for supporting the planning and scheduling of re-manufacturing activities of turbine blades, is presented.

Although robust scheduling approaches have the potential to bring significant advantages in the minimization of risks, it may lead to non-optimal solutions in a subset of scenarios. To overcome this limitation, proactive-reactive approaches [42] could serve and will be investigated in further research activities. Moreover, only the core repair process of the considered re-manufacturing process has been addressed in this paper. Future work will also address the extension to a wider portion of the whole re-manufacturing process, including inspection, disassembly, additional repair steps and re-assembly.

**Author Contributions:** The authors contributed equally to this work. All authors have read and agreed to the published version of the manuscript.

**Funding:** This research was partially funded by DIGIMAN4.0 project ("DIGItal MANufacturing Technologies for Zero-defect Industry 4.0 Production", http://www.digiman4-0.mek.dtu.dk/, accessed on 31 Decmber 2021). DIGIMAN4.0 is a European Training Network supported by Horizon 2020, the EU Framework Program for Research and Innovation (Project ID: 814225).

**Institutional Review Board Statement:** Not applicable.

**Informed Consent Statement:** Not applicable.

**Data Availability Statement:** Not applicable.

**Acknowledgments:** We thank Ansaldo Energia for the support in the definition of the requirements in relation to the planning and scheduling of remanufacturing activities for turbine blades.

**Conflicts of Interest:** The authors declare no conflict of interest.

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
