# Peer review of "A Robust Scheduling Framework for Re-Manufacturing Activities of Turbine Blades"

_applsci, doi:10.3390/app12063034_

Round 1

Reviewer 1 Report

Explain what is new and what this framework contributes to a better realization of re-manifesting activities
What are the advantages and disadvantages of the robust scheduling framework?
1. Line 24. Insert literature resources
2. In all article is missing reference
3. Could you explain Figure 4. and process phases
4. Considering shame in Figure 4, NDT is the last stage of the process. Is it necessary to do NDT upon removing damages in order to check the internal structure of damaged parts, and decide the level of defects ?????? If not, could you explain why it is not considered?
5. Gantt Chart is not visible and it is necessary to change the form or add it an addendum.
6. Conclusion is not clear. Could you explain the advantages of the proposed framework or disadvantages?

Reviewer 2 Report

  1. It has to be described why robust scheduling framework is better than other strategies, policies, and methods for re-manufacturing activities. In addition, what is the benefit of this method? (Abstract, Introduction, Example, or Conclusions)
  2. It seems that only one case study is provided to confirm the validity of the proposed method. The author should demonstrate the effectiveness of the proposed method in other examples in this paper.
  3. All references are cited in this manuscript as [?]. (Columns 19, 31, 37, 40, 43, 57, 60, 64, 66, 72, 76, 79, 81, 84, 86, 101….; Figures 1, 2, and 3)
  4. It seems that the Gantt chart in Figure 10 has no specific meaning and value in this research paper.
  5. For the integrity of the research paper, the reviewer suggests that some future work mentioned in conclusions needed to be done to extend this class of approaches to the whole re-manufacturing process of turbine blades.

Round 2

Reviewer 1 Report

Thank you for submitted changes and answers which improve your manuscript.

Reviewer 2 Report

I am satisfied with all replies from the authors.